# Sustainable Remediation of Soil and Water Utilizing Arbuscular Mycorrhizal Fungi: A Review

**DOI:** 10.3390/microorganisms12071255

**Published:** 2024-06-21

**Authors:** Xueqi Zhang, Zongcheng Wang, Yebin Lu, Jun Wei, Shiying Qi, Boran Wu, Shuiping Cheng

**Affiliations:** 1Key Laboratory of Yangtze River Water Environment, Ministry of Education, College of Environmental Science and Engineering, Tongji University, Shanghai 200092, China; 2111416@tongji.edu.cn (X.Z.); wangzongcheng66@163.com (Z.W.); boranwu@tongji.edu.cn (B.W.); 2Power China Huadong Engineering Corporation Limited, Hangzhou 311122, China; lu_yb@hdec.com (Y.L.); j@ecidi.com (J.W.); qshiying_7@163.com (S.Q.); 3Shanghai Institute of Pollution Control and Ecological Security, Shanghai 200092, China

**Keywords:** arbuscular mycorrhizal fungi, phytoremediation, heavy metal, organic pollutants, remediation mechanism, tolerance strategies

## Abstract

Phytoremediation is recognized as an environmentally friendly technique. However, the low biomass production, high time consumption, and exposure to combined toxic stress from contaminated media weaken the potential of phytoremediation. As a class of plant-beneficial microorganisms, arbuscular mycorrhizal fungi (AMF) can promote plant nutrient uptake, improve plant habitats, and regulate abiotic stresses, and the utilization of AMF to enhance phytoremediation is considered to be an effective way to enhance the remediation efficiency. In this paper, we searched 520 papers published during the period 2000–2023 on the topic of AMF-assisted phytoremediation from the Web of Science core collection database. We analyzed the author co-authorship, country, and keyword co-occurrence clustering by VOSviewer. We summarized the advances in research and proposed prospective studies on AMF-assisted phytoremediation. The bibliometric analyses showed that heavy metal, soil, stress tolerance, and growth promotion were the research hotspots. AMF–plant symbiosis has been used in water and soil in different scenarios for the remediation of heavy metal pollution and organic pollution, among others. The potential mechanisms of pollutant removal in which AMF are directly involved through hyphal exudate binding and stabilization, accumulation in their structures, and nutrient exchange with the host plant are highlighted. In addition, the tolerance strategies of AMF through influencing the subcellular distribution of contaminants as well as chemical form shifts, activation of plant defenses, and induction of differential gene expression in plants are presented. We proposed that future research should screen anaerobic-tolerant AMF strains, examine bacterial interactions with AMF, and utilize AMF for combined pollutant removal to accelerate practical applications.

## 1. Introduction

Soil and water are often the final destinations for chemical waste, and long-term pollution can threaten organisms in the food chain [1]. Human activity is a significant source of soil contamination, with waste from household, industry, and agriculture contributing to the problem [2]. When domestic or industrial wastewater is not properly treated, it can indirectly lead to soil pollution through activities such as irrigation [3]. Unfortunately, wastewater discharges from these activities and farming are common causes of water pollution [4]. Inadequate wastewater treatment in water treatment plants leads to the pollution of drinking water sources [5].

Phytoremediation is a sustainable approach that can be applied in situ, thereby avoiding secondary contamination [6]. The various methods of phytoremediation include phytovolatilization, phytoextraction, phytofiltration, phytostabilization, phytodegradation, and rhizodegradation. Among these methods, the rhizosphere of plants plays a crucial role in effective remediation [7]. Soil microorganisms, especially endophytes, contribute significantly to assisting plants in resisting biotic and abiotic stresses [8]. However, traditional phytoremediation methods have limitations such as low biomass production, high time consumption, and stress caused by the combined toxicity of contaminated media, which can diminish the remediation potential of plants [9]. Rational manipulation of the inter-root microbial assemblage is an effective measure to enhance phytoremediation efficiency and maintain sustainability.

Arbuscular mycorrhizal fungi (AMF) are important components of plant inter-root microbial communities [10]. They have shown the ability to improve plant growth in unfavorable environments and colonization of over 90% of angiosperms [11]. The exchange of nutrients between plants and AMF is the cornerstone for stabilizing this relationship [12]. Plants provide AMF with carbohydrates and lipids, while AMF utilize an extensive mycelial network to obtain water and mineral nutrients from the soil and supply them to the host plant through symbiotic interfaces [13]. The symbiosis between AMF and plants is considered to be one of the most effective ways to counteract biotic and abiotic stresses, including remediation of contaminated media and enhancement of plant tolerance [14,15]. 

AMF are found in various environments and play a crucial role in regulating ecosystem functions in polluted areas [16]. The concentration of pollutants in different environmental media affects the ecological composition of AMF. The discovery of pollutant-tolerant strains is essential for the study of AMF remediation of polluted media [17]. Studies have found that introducing indigenous AMF into contaminated soil promotes soil enzyme activities and antioxidant enzyme systems of host plants [18]. In contaminated areas, the symbiosis between AMF and plants creates favorable habitats for other soil microorganisms by providing a stable supply of organic material and stimulating root growth [19,20]. The increased soil organic matter resulting from the introduction of AMF traps more available toxic and harmful compounds, promoting the overall remediation of the polluted environment [21].

In this paper, we review the remediation ability, remediation strategy, and tolerance mechanism of the AMF–plant symbiosis system for different polluted environmental media. We outline prospective studies on the AMF–plant symbiosis system to provide new ideas and technologies for the reconstruction of vegetation in polluted areas and the management of polluted environments.

## 2. Bibliometric Analyses

Since the beginning of the 21st century, there has been a rapid increase in published research works on AMF-assisted phytoremediation, with the first article relevant to this topic published in 2000. A bibliometric analysis of the published literature in the last 23 years (2000–2023) was conducted to identify hotspot issues and associated important factors related to AMF-assisted phytoremediation. From the Web of Science (WOS) core collection database using the query TS = (“arbuscular mycorrhizal fungi”) AND TS = (“phytoremediation”), 520 relevant research articles published by researchers and scholars from around the world were retrieved. The number of annual publications and the percentage of annual publications during this period are shown in Figure 1. The number of published articles related to AMF-assisted phytoremediation consistently increased during this period. The year 2021 had the highest number of publications, with 56 articles, accounting for 11% of the total (Figure 1A). The retrieved 520 articles were analyzed using VOSviewer1.6.18. According to the co-author analysis results, there are 35 authors with 5 or more publications. Among them, Guo Wei has the highest number of publications (13) (Figure 1B). In addition, the country with the highest number of publications was China (187), followed by India (35), the United States (34), Brazil (32), Canada (32), and France (32), which indicate that research on AMF-assisted phytoremediation mainly originated from China (Figure 1C).

Co-occurrence clustering of all keywords was also analyzed by VOSviewer. The minimum number of occurrences of terms was defined as 10, and irrelevant and duplicate terms were merged. After these specific selections, keyword co-occurrence network graphs were generated. The keyword co-occurrence network diagram generated based on the above settings is shown in Figure 2, and the keyword co-occurrence network diagram is divided into 6 clusters. The terminology map shows the hotspot issues and important factors related to AMF-assisted phytoremediation. Among them, the three clusters with the highest term frequency are labeled in red, purple, and green. The red group indicates that the contaminated environmental medium in which AMF-assisted phytoremediation is applied is mainly soil. The green group indicates that promotion of plant growth is an important part of the AMF-assisted phytoremediation process. The purple group indicates that heavy metal (HM) is the pollutant species with the highest frequency of AMF-assisted phytoremediation being applied. The dark blue group makes it clear that the rhizosphere is the primary site of AMF action, and it is closely associated with rhizospheric microbial communities. Therefore, we summarized the research advances from these four clusters below.

## 3. AMF-Assisted Phytoremediation of Contaminated Soil

### 3.1. Soil Contaminated with Heavy Metal

HMs can negatively impact the morpho-physiological and biological properties of plants [22]. They even extend to the food chain and cause harm to human health [23]. While some HM-hyperaccumulating plants play a crucial role in phytoremediation, this phenomenon is rare in terrestrial higher plants [24]. Therefore, the efficient phytoremediation of soil contaminated with HMs requires the manipulation of microbial and plant combination [25]. HMs are the most widespread soil contaminants for AMF remediation application. It has been reported in 1990 that inoculation of AMF in soil with either too low or too high concentrations of HMs can ameliorate both of these soil problems [26]. Although the spore diversity of AMF in soil contaminated with HMs tends to be lower than those in uncontaminated soil, this does not mean that the beneficial effects of AMF will be limited [27]. Silva-Castro et al. (2023) demonstrated that indigenous AMF-colonized plants exhibited higher levels of HM accumulation compared to non-indigenous AMF-colonized plants [18]. Therefore, some AMF genera screened from contaminated areas may be highly tolerant to HMs.

Among the studies utilizing AMF for the remediation of soil contaminated with HMs, most attention has been on Zn, Cd, Cu, and Pb. A study by Cabral et al. (2010) reported that the retention of several genera of AMF showed the order of Cu > Zn > Cd > Pb and that soil trace elements, such as Cu and Zn, were retained in AMF tissues at a rapid rate [28]. The inoculation of AMF into plants grown in HM-contaminated areas was able to increase the length of the root system of the host plant and significantly improve the contact area of the plant with the contaminated soil [29]. AMF-colonized plants showed higher root HM content and lower HM content in shoots and fruits compared to non-mycorrhizal plants [14]. These findings demonstrate that AMF in soil contaminated with HMs can reduce HM concentrations and protect plants from HM toxicity.

### 3.2. Soil Contaminated with Organic Pollutants

AMF lack the enzymes to directly break down organic matter, but their colonization in contaminated soil is significantly higher than in uncontaminated soil [30]. In most of the organically contaminated sites where AMF are present, AMF promote the growth and development of host plants to enhance the uptake of organic pollutants while mitigating the hazards of pollution toxicity [31]. Additionally, AMF stimulate the exudation of plant root secretions, enhancing the uptake and degradation of pollutants by microorganisms, such as glomalin [32]. Overall, AMF play an important role in promoting the degradation and removal of organic pollutants from contaminated soil.

#### 3.2.1. Petroleum Hydrocarbons (PHCs) and Polycyclic Aromatic Hydrocarbons (PAHs)

Industrial activities have led to PHCs being a kind of common organic pollutant in soil [32]. PHCs enter the ecosystem through leaking underground storage tanks, oil spills, transportation processes, and industrial activities. The extensive contamination of soil by PHCs makes it extremely hydrophobic and infertile, ultimately resulting in a reduction in plant and microbial biomass [33]. The total petroleum hydrocarbon (TPH) concentration influences the AMF community structure, which, in turn, may impact bacteria and fungi associated with AMF spores [34]. However, AMF could be identified in plant roots or inter-plant root soil even when there was a high level of TPH contamination. The contamination levels were as high as 91,000 µg TPH/kg of soil, equivalent to 9.1% (*w*/*w*) of the TPH content in the soil [35]. This suggests that certain AMF species that are tolerant to high concentrations of TPH may exist in areas that are naturally contaminated with TPH. To enhance the efficiency of TPH degradation, Alarcón et al. (2008) inoculated AMF with a blend of petroleum-degrading microorganisms and found that the degradation efficiency increased in the rhizosphere layer significantly [36].

TPH contains aliphatic and aromatic compounds. Among the aromatic compounds, PAHs are considered risky due to their toxicity, mutagenicity, and carcinogenic properties. PAHs are produced from various sources such as transportation emissions, residential gas, and industrial production. These PAHs are widespread due to their diverse sources and semi-volatility, making them ubiquitous [37]. Plant growth and PAH uptake, along with inter-root soil bioactivity induced by the AMF–plant symbiotic system, significantly reduced the residual amount of PAHs in soil [38]. AMF colonization was able to increase the PAH concentration in host plant root and aboveground parts by more than 50% in PAH-contaminated soil [39].

#### 3.2.2. Polychlorinated Biphenyls (PCBs)

In 2004, the Stockholm Convention identified PCBs as one of the 26 persistent organic pollutants (POPs) that needed to be reduced globally [40]. Over 90 countries have since committed to eliminating and disposing of large quantities of PCBs. However, PCBs remain a significant threat to ecosystems due to their persistence in the environment and slow decomposition. Qin et al. (2016) discovered that AMF hyphae could speed up the dissipation of PCB congeners in soil, and there was a significant correlation between the hyphal length of AMF and the rate of PCB dissipation [41]. Before this study, other research studies used AMF in combination with other helpful organisms to treat PCB-contaminated soil. Lu et al. (2014) found that earthworms and AMF colonization not only increased the biomass production of the host plant but also resulted in the accumulation of PCBs in the plant [42]. Combined application of AMF with beneficial bacteria yields more significant results. Alfalfa shoots inoculated with both AMF and rhizobium accumulated higher concentrations of PCBs than those inoculated with only rhizobium [43]. These studies all suggest that the main reason for the reduction in PCB concentration in soil is the microbiota benefiting from the mycorrhizal binding process. This speculation was confirmed by Qin et al. (2014): inoculation with AMF increased the proportion of Alphaproteo bacteria in the rhizosphere and significantly correlated with the dissipation rate of PCBs in soil [44].

In recent years, novel materials have been employed to break down PCBs in soil. Nanoscale zero-valent iron (nZVI) is highly effective in degrading PCBs in soil but may pose a threat to plants when used in conjunction with phytoremediation [45,46,47]. This will undoubtedly limit the efficiency of nZVI for PCB degradation. Fortunately, AMF can alleviate nZVI-induced toxicity in plants [48]. But a high concentration (10 g/kg) of nZVI in soil inhibits AMF diversity. However, this restriction does not work in the medium-concentration (1 g/kg) soil but rather promotes the remediation of PCB-contaminated soil [49]. This suggests that the co-application of AMF with nanomaterials shows great potential for application in PCB-contaminated soil remediation.

#### 3.2.3. Antibiotics

Animal husbandry, agriculture, and aquaculture are the primary sources of antibiotics found in soil [50]. A significant amount of antibiotics cannot be metabolized by animals and are released into nature through animal manure fertilization and wastewater [51]. Antibiotics are present in soil at concentrations ranging from 0.7 µg/kg to 5 g/kg [52]. The adsorption of antibiotics to soil increases their persistence in the environment, which, in turn, can lead to antibiotic resistance [53]. The addition of AMF increased soil enzyme activities and AMF released secretions containing assimilates into the rhizosphere of the host plant, which led to an increase in soil microbial biomass [54]. Glomalin, which is a mycelial secretion produced by AMF, may stimulate the increase in microflora with oxytetracycline (OTC) degradation capacity [55]. Furthermore, AMF have the ability to enhance the removal efficiency of soil OTC by promoting host plant growth and development through improving the soil nitrogen content [56].

## 4. AMF-Assisted Phytoremediation of Contaminated Water

In both natural wetlands and constructed wetlands, microorganisms play a crucial role in maintaining the ecosystem’s health [57]. However, excess water may negatively affect AMF, as these microorganisms are sensitive to a lack of oxygen [58]. The discovery of AMF in some wetlands has provided new idea for the remediation of polluted waters [59].

Earlier research has demonstrated that there is a high presence of AMF in ecological floating beds (EFBs) that are perpetually flooded despite the absence of any external introduction of these fungi [60]. This indicates that using AMF in water treatment systems that rely on plant-based processes could be highly beneficial. However, the actual advantages of these AMF communities are not yet fully understood. Several studies have applied AMF-assisted phytoremediation in various human-intervened water treatment systems, such as constructed wetland (CW), EFBs, and stormwater filters, confirming earlier speculations (Table 1). In such installations, AMF have been used to increase the pollutant extraction efficiency of plants, resulting in an overall improvement of the system’s pollutant removal capability [61]. To improve AMF colonization in aquatic habitats, researchers have been exploring various inoculation methods. In some studies, AMF–plant symbiosis systems pre-cultivated in pots were transplanted into water treatment systems [62,63]. In most of the remaining studies, the AMF inoculum was either spread on the surface or middle layer of the substrate of the water treatment system or mixed with it [64,65,66,67]. HMs and nutrients are the most commonly treated contaminants using AMF in water treatment systems. Furthermore, AMF have also been utilized in the treatment of emerging contaminants in contaminated water.

## 5. Direct Influence of AMF on Pollutant Removal

### 5.1. Binding and Stabilization of Pollutants by Glomalin

Glomalin-related soil protein (GRSP) is a glycoprotein produced by AMF spores and mycelial walls [74]. It can remain in the soil for up to 42 years and has the potential to maintain soil health with long-term sustainability [75]. GRSP is considered one of the key mechanisms for AMF-assisted phytoremediation of contaminated soil. Studies on rhizosphere and non-rhizosphere soil of mycorrhizal plants in Cu-contaminated ecosystems have found that GRSP-Cu is one of the main forms of “immobilized” Cu in contaminated soil [76,77]. In contaminated soil, GRSP binds to HMs mainly through functional groups such as carboxyl (-COO-), hydroxyl (-OH), carbonyl (-C=O), and amide (-CO-NH) to reduce the toxicity and bioavailability of HMs [78]. This binding effect enables GRSP to be released in rhizosphere soil to aggregate HMs in the roots, inhibiting their transfer into the aerial parts and reducing damage [79].

Soil aggregates have a large surface area and many adsorption sites, making them highly enriched for most HMs [80]. GRSP plays an important role in forming soil macroaggregates. Li et al. (2022b) found that in soil contaminated with Pb, the content of GRSP is significantly and positively correlated with the stability of soil aggregates [81]. Soil aggregation and structural stability can affect soil organic carbon (SOC) sequestration. Long-term HM contamination can increase Basidiomycete, which promotes carbon degradation, and decrease Proteobacteria, which promotes carbon fixation. This leads to SOC loss, affecting SOC storage [82]. At a HM-contaminated site, it was found that GRSP carbon contributed 89% of the total SOC, suggesting the potential of GRSP to sequester SOC under extreme stress [83]. In addition, GRSP can also serve as an additional carbon source and catabolic substrate for microorganisms, which improves microbial catabolic function by promoting the release of extracellular enzymes [84] (Figure 3).

### 5.2. Pollutant Retention inside Fungal Structure

The most significant aspect of AMF is their mycelium network which spreads beyond the root and is present everywhere, increasing the ground contact area between the plant root system and contaminated soil. The AMF mycelium can store HMs at 10–20 times more than the plant root [85]. It overcomes one of the significant limitations of phytoremediation [32]. In mycorrhizal plants, AMF colonization of plant roots is limited to epidermal and cortical tissues [86]. Wu et al. (2016) used synchrotron radiation micro-X-Ray Fluorescence (SR-μXRF) to observe the exposure of dandelion (*Taraxacum platypecidum*) to Cr and showed that AMF colonization inhibited the transport of Cr from roots to shoots [87]. Moreover, Cr was present in both the cortex and vascular bundles in the main roots, which was not colonized by AMF, while Cr was detected only in the cortex of mycorrhizal main roots [88]. This confirms the ability of AMF colonization to prevent the transfer of HMs to plant transport tissues and avoid damage to the photosynthetic apparatus (Figure 4). HMs are transported through extraradical mycelium, and in mycorrhizal plants, metal ions mainly accumulate in the fungal structures within the root’s cortical cells [89]. Alvarado-López et al. (2019) discovered that Pb treatment was capable of inducing AMF colonization within the roots of carrots (*Daucus carota*), resulting in a greater percentage increase in the number of vesicles and mycelium [90]. The adsorption and filtration effects within intraradical mycelium can limit the movement of HM ions into plant roots’ cytoplasm [91]. Meanwhile, AMF vesicles can accumulate a large amount of HMs when subjected to HM stress [92]. Salazar et al. (2018) observed AMF spores in Pb-contaminated soil using micro-X-ray Fluorescence (μXRF) and found that AMF spores functioned in a similar way to vesicles and had an accumulative effect on Pb [93]. 

The process of HM immobilization in AMF fungal structures is accompanied by the occurrence of complexation reactions, which are essential for reducing damage to mycorrhizal plants. Wu et al. (2016) found that fungal structures and fungal cell walls within mycorrhizal roots were able to provide a site for the reduction of Cr (VI) to Cr (III), and complexation via carboxylate (-COOH) ligands or histidine analogues of Cr (III), thereby inhibiting the transfer of Cr to the plant root cytoplasm [87]. Thus, the retention of HMs by mycorrhizal plants increased with increasing concentrations in the soil, and the formation of arbuscules and vesicles increased. HMs are immobilized in fungal hyphae associated with mycorrhizal plants, which retains HM ions in spores, vesicles, and arbuscules through mycelium transport, and prevents the transfer of HMs to the plant cytoplasm through complexation reactions. This is one of the tolerance strategies through which AMF are directly involved in providing a physical barrier to prevent HMs from entering the plant.

### 5.3. Trophic Interaction between AMF and Host Plant

Previous research has shown that communities of AMF can allocate more nutrients to plant roots than to the surrounding soil and minimize the damage to plants caused by external disturbances [94]. These fungi are specialized symbionts that use host plants as their sole source of carbon, relying on plant fatty acids to complete their asexual life cycle [95]. They are estimated to receive 4–20% of the total carbon fixed through photosynthesis. In return, AMF provide plants with nitrogen (N), phosphorus (P), and other nutrients, which are usually more difficult to obtain in contaminated soil [32]. The study by Huang et al. (2020) has shown that the supply of P is more crucial than that of N in restoring plant communities in soil contaminated with HMs [96]. In the plant cell, P is a key element for several physiological and biochemical functions. Improving P uptake through AMF is an important mechanism that can benefit mycorrhizal plants that are growing in soil with high concentrations of HMs [97]. AMF promote P uptake in plants by increasing P acquisition by the mycelial network, promoting P mobility in soil, and increasing the activities of P and phytase enzymes [98]. The acquisition of inorganic P at the mycorrhizal symbiotic interface accounted for 70–100% of the plant’s total pi acquisition [99,100]. You et al. (2022) found that the efficiency of AMF in assisting plants to remediate soil contaminated with Cd was affected by the concentration of P [101]. The addition of P allowed AMF to induce the entry of Cd through the Fe uptake pathway, which facilitated the uptake of Cd by the plants. Changes in the level of phosphorus led to the diversification of the mechanisms through which AMF regulated Cd uptake.

## 6. The Role of AMF in Host Plant’s Tolerance Strategy

### 6.1. The Role of AMF in Host Plant’s Tolerance Strategy

Plants’ tolerance to HMs and their detoxification mechanisms are related to the subcellular distribution of HMs and their chemical forms [102]. A significant portion of cellular HMs are sequestered in the vacuoles of cortical cells, which can be in the form of free ions or bound to specific molecules like phytochelatins (PCs). Upon exposure to HMs, PCs are synthesized in plants using glutathione (GSH) as a precursor. The HM-PC complex is then transported into a vesicle via the ATP-Binding Cassette (ABC) transporter [103]. This process of sequestration is also present in the cell wall. AMF colonization increases the amount of cell wall polysaccharides (e.g., hemicellulose and pectin), which help to bind HMs to the host plant’s cell wall [104]. Han et al. (2021) found that AMF-colonized ryegrass sequesters more Cd in the root cell wall and vesicles under Cd stress compared to other organelles [105]. This sequestration effect is seen more prominently with elevated Cd concentrations. Gao et al. (2021) reported similar findings, showing that Cd is captured by the cell wall and primarily stored in hemicellulose and pectin components [106].

The biological functions of HMs are closely related to their chemical forms [107]. Different chemical forms of Cd have varying degrees of toxicity. Colonization by AMF can convert HMs into forms that are less toxic and inactive to plants. Zhang et al. (2019) confirmed that inoculation with AMF significantly reduced the proportion of inorganic and water-soluble forms of Cd (which cause the greatest Cd stress to plants) in the shoot and root of maize (*Zea mays*) [108], while it increased the proportion of pectate acid and protein-bound insoluble phosphate forms of Cd. Similar results were reported in [109].

### 6.2. AMF Response to Host Plant’s Antioxidant Defense

One of the ways pollutants affect plants is by directly or indirectly producing an excess of reactive oxygen species (ROS) [110]. A high level of ROS can impact the redox state of plant cells and cause oxidative damages [111]. Plants develop both enzymatic and non-enzymatic antioxidant defense mechanisms to eliminate excess ROS and ensure their survival [112]. When excess pollutants are transported to the photosynthetic organs of mycorrhizal plants, AMF promote the activities of antioxidant enzymes such as superoxide dismutase (SOD), peroxidase (POD), catalase (CAT), ascorbate peroxidase (APX), glutathione peroxidase (GPX), and glutathione S-transferase (GST), which protect plants from the damage caused by these pollutants [113,114,115]. On the other hand, regulation of the ascorbate–glutathione (ASA-GSH) cycle in the cytoplasm and chloroplasts by AMF enhances plant tolerance to HM stress, which resists oxidative stress and scavenges reactive oxygen radicals to maintain normal photosynthesis [116] (Figure 5).

Plants that are exposed to abiotic stress often produce toxic aldehydes, such as methylglyoxal (MG). The glyoxalase system, which consists of glyoxalase I (Gly I) and glyoxalase II (Gly II) in combination with GSH, is responsible for the detoxification of MG [117]. Li et al. (2022a) found that AMF can alleviate the negative effects of Cd stress on wheat by speeding up the ASA-GSH cycle [118]. AMF increase the activity of APX, which reduces the production of ROS and oxidative damage by promoting the production of GSH and metallothioneins (MTs) and by breaking down MG. The increased production of GSH and MTs helps in trapping and immobilizing Cd, hence reducing its toxicity [119].

### 6.3. AMF Regulate Gene Expression Resistance to Pollutant Stress 

AMF indirectly affect the tolerance strategies of host plants in two ways, as described above. These processes are successful due to changes in the expression of various transcribed genes involved in the host plant’s antioxidant defense system, transporter activity, and cell wall biosynthesis. 

Antioxidant enzymes play a crucial role in removing harmful ROS in living organisms [120]. Ban et al. (2023) found that under stress from CuO-NPs, AMF can up-regulate the expression of six genes that encode GST and three genes that encode POD in plants [68]. This suggests that AMF can help plants detoxify ROS and protect them against oxidative damage. Similarly, Wang et al. (2022) reported that AMF colonization increased the expression of genes that encode antioxidant enzymes, including *MtSOD*, *MtPODp7*, and *MtCAT4*, in Medicago truncatula leaves under Cd stress [121]. This led to an increase in ROS scavenging and a decrease in oxidative damage caused by Cd exposure.

ABC transporters are a class of transmembrane transport proteins commonly found in prokaryotes and eukaryotes, which have been shown to transport As, Cd, and Hg [122]. In plants, ABC transporter proteins are present in the membranes of chloroplasts, mitochondria, peroxisomes, and vesicles. They play vital roles in maintaining cellular osmotic homeostasis, toxin enrichment and efflux, and stomatal motility in response to different types of stress [123]. The genes encoding the ABC transporter family were significantly down-regulated when the accumulation of Cd in the root system of AMF-colonized plants reached the triggering threshold of homeostasis [124]. This strategy can help enhance tolerance and establish good growth under Cd stress via the symbiotic relationship between AMF and plants. The extraradical mycelium of AMF can increase the scope for soil foraging and Pi (phosphate) uptake by extending beyond the Pi depletion zone. Molecular analyses have confirmed that specific members of plants’ PHT1 Pi transporter family play a role in this Pi uptake pathway [125]. The P transporter gene *MtPT4* of *M.truncatula* is localized in the periplasm of tufts and is considered to be a marker of symbiotic function. *MtPT4* encodes phosphate transporter proteins involved in the transfer of Pi from AMF to the plant as required [126]. Watts-Williams et al. (2017) reported that as the Zn concentration in the soil increased, AMF induced an increase in the expression of the P transporter gene *MtPT4* [127]. This allowed the maintenance of AMF colonization while increasing the Zn concentration in plant biomass and decreasing it in tissue. Li et al. (2018) reported similar results for *Medicago sativa* under As stress. AMF up-regulated the gene *MsPT4*, thus improving plant phosphorus levels [128].

Maintaining the stability of cell membranes and cell walls is a key strategy used by plants to cope with abiotic stress [129]. Phosphatidylcholine is responsible for maintaining the integrity of cell membranes in higher plants and plays a significant role in helping plants adapt to environmental stresses [130]. Cui et al. (2022) demonstrated that genes encoding phospho-base N-methyltransferase 3 (PMT3) catalyses, which are responsible for synthesizing phosphatidylcholine, were significantly up-regulated in AMF-colonized Suaeda salsa plants when subjected to the combined stress of Cd and salt [131]. Furthermore, the cell wall serves as the first line of defense against various adverse abiotic and biotic environmental influences. AMF colonization significantly enhances the expression of genes involved in the synthesis of cell wall organization when subjected to abiotic stress [132]. Zhang et al. (2021) reported that AMF colonization caused Pb-stressed plants to increase the transcription of *MtPrx05* and *MtPrx10* in the root system by 65% and 148%, respectively [133]. These two genes were hypothesized to be involved in polysaccharide cross-linking and cell wall hardening.

## 7. Conclusions and Perspectives

Over the past 23 years, breakthroughs have been made in studies on AMF-associated phytoremediation. Studies showed that AMF could establish symbiotic relationships with a wide range of plant species and effectively remediate contaminated soil and water. When exposed to pollution, AMF secrete GRSP, retain pollutants inside their fungal structure, and enhance the plant biomass for the removal of pollutants. Furthermore, AMF promote the immobilization of pollutants in a non-toxic form in the root subcellular structures, respond to the plant defense system, and regulate several functional protein genes to protect plants from pollutant toxicity. However, the practical applications of AMF still need to overcome some serious challenges. Future research may focus on the following areas:
The lack of oxygen in water and sediment is one of the factors that limit the effectiveness of AMF. To overcome this, it is important to study and identify AMF strains that can adapt to anaerobic environments and have a wide range of applications. Additionally, it is essential to optimize the combination of AMF strains with plant species to maximize the efficiency of remediation.Pollution can affect the community structure of AMF applied to contaminated sites. Only a few AMF species with resistance to pollutant stress can survive in highly polluted areas, which can diminish the expected remediation effect. Future studies focusing on screening and isolating bacterial species that closely interact with AMF in highly polluted environments and understanding the interactions between them should be conducted. Developing strategies that utilize AMF and microbial communities, which interact with AMF, rather than inoculation with AMF alone, is essential for the remediation of contaminated soil and water. Existing research has focused on the remediation of a single type of pollution via AMF–plant symbiosis, while the interactions between different types of pollutants have been neglected. Remediation of composite polluted media with HMs and organic compounds should be emphasized in the future. To address the above challenges, medium-scale remediation trials of actual pollution and site construction should be conducted to accelerate the transition of AMF–plant symbiosis system remediation technology from simulation trials to practical applications.

## Figures and Tables

**Figure 1 microorganisms-12-01255-f001:**
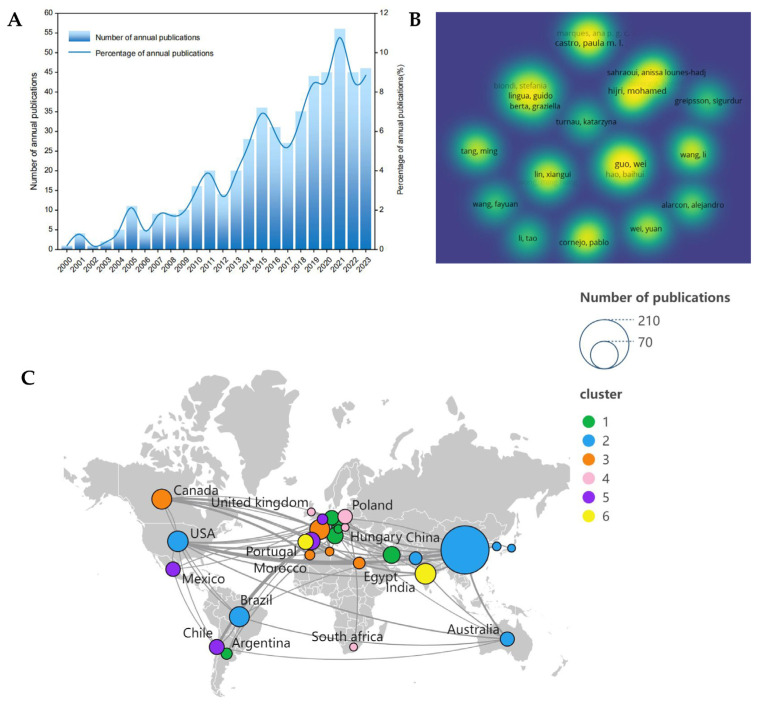
(**A**) Annual total number and percentage of publications based on the WOS core collection database from 2000 to 2023. (**B**) Co-authorship analysis of authors with ≥5 publications. (**C**) Co-authorship clustering analysis of countries.

**Figure 2 microorganisms-12-01255-f002:**
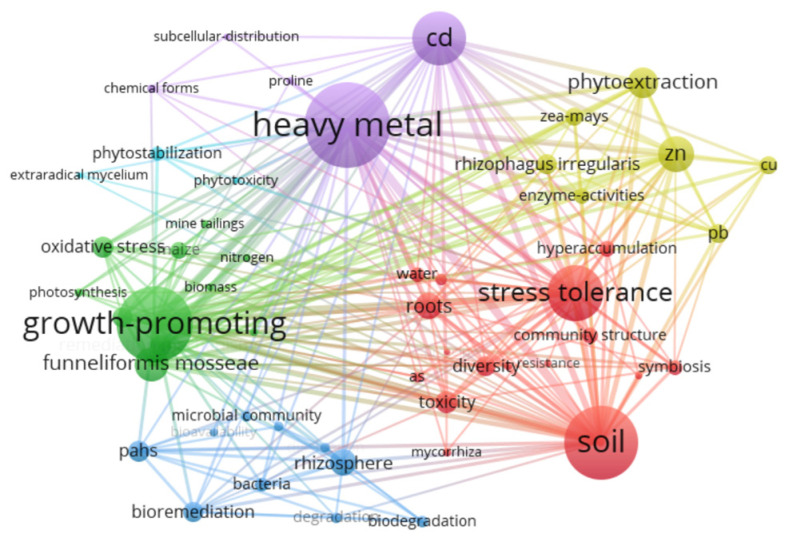
Hotspot cluster analysis of keywords. Different colors indicate different clusters.

**Figure 3 microorganisms-12-01255-f003:**
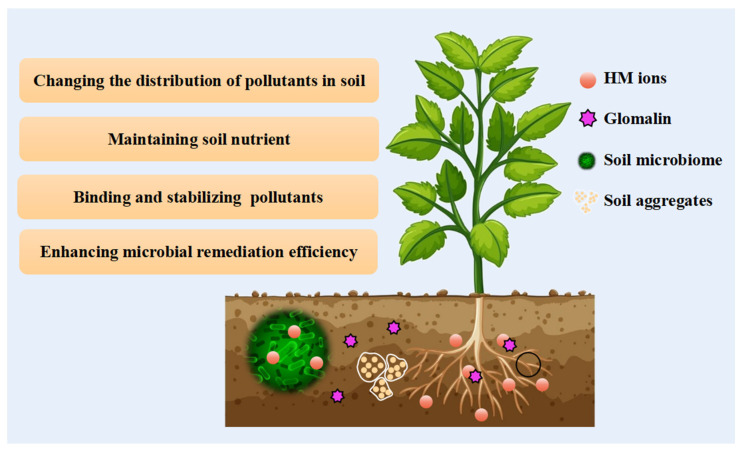
Different beneficial mechanisms of glomalin-mediated remediation of contaminated soil.

**Figure 4 microorganisms-12-01255-f004:**
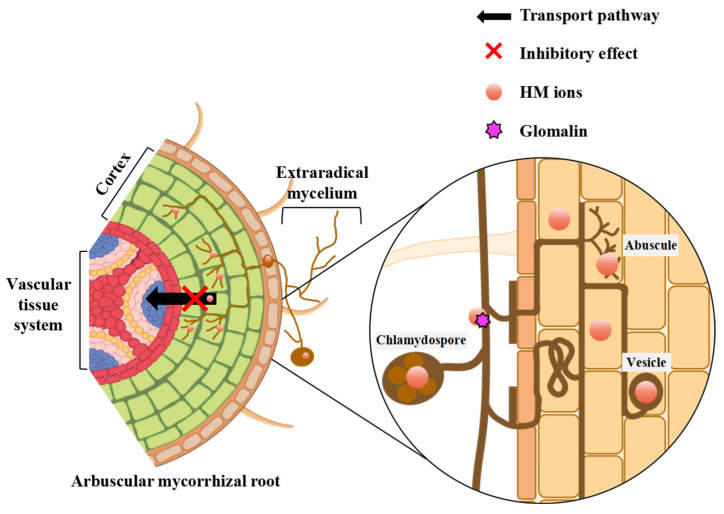
The mechanism of AMF-induced heavy metal ion retention inside their fungal structures.

**Figure 5 microorganisms-12-01255-f005:**
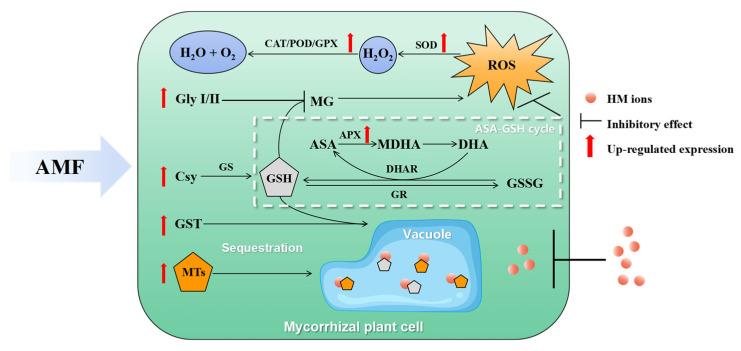
Antioxidant defense mechanisms of AMF-induced host plants to tolerance to HM stress.

**Table 1 microorganisms-12-01255-t001:** Different AMF–aquatic plant symbiosis systems are utilized in various types of wastewater treatment systems.

AMF Species	Water Treatment Systems	Aquatic Plants	Pollutants	Reference
*Funneliformis mosseae*	Vertical Flow Constructed Wetland (VFCW)	*Phragmites australis*	Pb, Zn, Cu, and Cd	[65]
*F. mosseae*	VFCW	*P. australis*	Copper oxide nanoparticles (CuO-NPs)	[68]
*F. mosseae*	VFCW	*P. australis*	COD, TN, and CuO-NPs	[62]
*F. mosseae*	VFCW	*Canna indica*	Tetracycline and Cu	[66]
*F. mosseae*	VFCW	*C. indica*	Sulfamethoxazole, Cu, and Cd	[69]
*F. mosseae*	VFCW	*Pteris vittata*	As	[63]
*F. mosseae*	EFB	*Zea mays*	Pb	[70]
*Glomus etunicatum*	EFB	*Cyperus alternifolius*	TDS, COD, TN, TP, and salt ions	[71]
*Rhizophagus irregularis*	VFCW	*Glyceria maxima*	PPCPs (Ibuprofen and diclofenac)	[67]
*R. irregularis*	VFCW	*G. maxima*	PPCPs (Hydrochlorothiazide, chloramphenicol, furosemide, gemfibrozil triclosan and triclocarban)	[64]
Commercially available mycorrhizal inoculant	Stormwater biofilter	*Ficinia nodosa* *Juncus australis* *Carex appressa*	TN, TP, phosphate, and Cd	[72]
Native AMF communities	Horizontal Subsurface Flow Constructed Wetland (HFCW)	*Canna flaccida* *C. indica* *Watsonia borbonica* *Agapanthus africanus* *Zantedeschia aethiopica*	TSS, BOD_5_, COD, PO_4_^3−^, and NH_4_^+^	[73]

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
