# Peer review of "Sustainable Remediation of Soil and Water Utilizing Arbuscular Mycorrhizal Fungi: A Review"

_microorganisms, 2024, doi:10.3390/microorganisms12071255_

Round 1

Reviewer 1 Report

Comments and Suggestions for Authors

The study is interesting and in line with the journal, however some details can be improved for better clarity and quality.

Abstract: is necessary to include the period of the studies analyzed for the bibliometric analysis and to include the total quantity of studies identified.

Bibliometric analysis: line 82. To explain why the investigation was analyzed in the period 2000-2023?

In all the manuscript is necessary to check the spaces between text and brackets or between text and parentheses.

In the figure 2, is important to describe about the blue group

After the results obtained in the figure 2, is important to put a map about researcher collaboration co-occurrence about the topic in study period around the world and a Network visualization of the main countries in such investigation.

In table 1 change BOD5 by BOD5

In table 1, VFCW is defined in the firs line, but EFBs is not defined

I suggest including a word cloud with the main benefits of arbuscular mycorrhizal fungi in the database obtained

Reviewer 2 Report

Comments and Suggestions for Authors

The article "Sustainable remediation of soil and waters utilizing arbuscular mycorrhizal fungi: A review" provides a comprehensive review of the role of plant commensal AMF. The authors have put excellent effort into gathering a lot of information and presenting it in a structured way in this manuscript. In terms of corrective actions, only a few minor grammatical formatting is required. The authors should mention the versatile characteristics of AMF in surviving on different soil types and climatic conditions.

Comments on the Quality of English Language

Grammatical formatting and minor modifications are required.

Reviewer 3 Report

Comments and Suggestions for Authors

The manuscript: Sustainable remediation of soil and waters utilizing arbuscular mycorrhizal fungi: A review presents the published literatures on the topic arbuscular mycorrhizal fungi phytoremediation; thus, the article is suitable for the journal. A bibliometric analysis of the published literature in the last 23 years (2000-2023) was conducted. In general, the manuscript is well written, the literature is clearly presented, and this can be accepted after minor revisions:

  1. Please add a space between before the brackets with the references  
  2. The authors should states why they choose the last 23 years for analysis
